# Neural Graph Machines: Learning Neural Networks Using Graphs

**Thang D. Bui**[*]
University of Cambridge
tdb40@cam.ac.uk

**Sujith Ravi**
Google Research
sravi@google.com

**Vivek Ramavajjala**
Google Research
vramavaj@google.com

## Abstract

Label propagation is a powerful and flexible semi-supervised learning technique on graphs. Neural network architectures, on the other hand, have proven track records in many supervised learning tasks. In this work, we propose a training objective for neural networks, *Neural Graph Machines*, for combining the power of neural networks and label propagation. The new objective allows the neural networks to harness both labeled and unlabeled data by: (a) allowing the network to train using labeled data as in the supervised setting, (b) biasing the network to learn similar hidden representations for neighboring nodes on a graph, in the same vein as label propagation. Such architectures with the proposed objective can be trained efficiently using stochastic gradient descent and scaled to large graphs. The proposed method is experimentally validated on a wide range of tasks (multi-label classification on social graphs, news categorization and semantic intent classification) using different architectures (NNs, CNNs, and LSTM RNNs).

## 1 Introduction

Semi-supervised learning is a powerful machine learning paradigm that can improve the prediction performance compared to techniques that use only labeled data, by leveraging a large amount of unlabeled data. The need of semi-supervised learning arises in many problems in computer vision, natural language processing or social networks, in which getting labeled datapoints is expensive or unlabeled data is abundant and readily available.

There exist a plethora of semi-supervised learning methods. The simplest one uses bootstrapping techniques to generate pseudo-labels for unlabeled data generated from a system trained on labeled data. However, this suffers from label error feedbacks (Lee, 2013). In a similar vein, autoencoder based methods often need to rely on a two-stage approach: train an autoencoder using unlabeled data to generate an embedding mapping, and use the learnt embeddings for prediction. In practice, this procedure is often costly and inaccurate in practice. Another example is transductive SVMs (Joachims, 1999), which is too computationally expensive to be used for large datasets. Methods that are based on generative models and amortized variational inference (Kingma et al., 2014) can work well for images and videos, but it is not immediately clear on how to extend such techniques to handle sparse and multi-modal inputs or graphs over the inputs. In contrast to the methods above, graph-based techniques such as label propagation (Zhu & Ghahramani; Bengio et al., 2006) often provide a versatile, scalable, and yet effective solution to a wide range of problems. These methods construct a smooth graph over the unlabeled and labeled data. Graphs are also often a natural way to describe the relationships between nodes, such as similarities between embeddings, phrases or images, or connections between entities on the web or relations in a social network. Edges in the graph connect semantically similar nodes or datapoints, and if present, edge weights reflect how strong such similarities are. By providing a set of labeled nodes, such techniques iteratively refine the node labels by aggregating information from neighbours and propagate these labels to the nodes' neighbours. In practice, these methods often converge quickly and can be scaled to large datasets with a large label space (Ravi & Diao, 2016). We build upon the principle behind label propagation for our method.

---

[*]Work done during an internship at Google

Another key motivation of our work is the recent advances in neural networks and their performance on a wide variety of supervised learning tasks such as image and speech recognition or sequence-to-sequence learning (Krizhevsky et al., 2012; Hinton et al., 2012; Sutskever et al., 2014). Such results are however conditioned on training very large networks on large datasets, which may need millions of labeled training input-output pairs. This begs the question: can we harness previous state-of-the-art semi-supervised learning techniques, to jointly train neural networks using limited labeled data and unlabeled data to improve its performance?

**Contributions:** We propose a discriminative training objective for neural networks with graph augmentation, that can be trained with gradient descent and efficiently scaled to large graphs. In particular, we introduce a regularization term for generic neural network architectures that enforces similarity between nodes in the graphs. This is inspired by the objective function of label propagation. The resulting cost is amenable to stochastic training and can be applied to various model classes. We also investigate using graphs as direct inputs to train neural network classifiers and experimentally demonstrate that this procedure is more efficient and accurate than previous two-stage approaches such as finding embeddings and using them for classification.

The closet approach to our work is the framework proposed by Weston et al. (2012), we extend their work in several ways: (a) our proposed training scheme is flexible, for example multiple graphs from multiple domains can be combined, (b) we provide extensive experiments on different types of neural networks and on properly constructed graphs (in contrast to *nearest neighbor* graphs in Weston et al. (2012), (c) we propose using graphs as inputs to the neural networks if there are no input features. Our work is also different from recent works on using neural networks *on* graphs (e.g. see Niepert et al. (2016)). Instead, we advocate a training objective that *uses* graphs to augment neural network learning.

## 2 BACKGROUND

In this section, we will lay out the groundwork for our proposed training objective in section 3.

### 2.1 LABEL PROPAGATION

We first provide a concise introduction to label propagation and its training objective. Suppose we are given a graph $G = (V, E, W)$ where $V$ is the set of nodes, $E$ the set of nodes and $W$ the edge weight matrix. Let $V_l, V_u$ be the labeled and unlabeled nodes in the graph. The goal is to predict a soft assignment of labels for each node in the graph, $\hat{Y}$, given the training label distribution for the seed nodes, $Y$. Mathematically, label propagation performs minimization of the following convex objective function, for $L$ labels,

$$\mathcal{C}_{\text{LP}}(\hat{Y}) = \mu_1 \sum_{v \in V_l} \left\| \hat{Y}_v - Y_v \right\|_2^2 + \mu_2 \sum_{v \in V, u \in \mathcal{N}(v)} w_{u,v} \left\| \hat{Y}_v - \hat{Y}_u \right\|_2^2 + \mu_3 \sum_{v \in V} \left\| \hat{Y}_v - U \right\|_2^2, \quad (1)$$

subject to $\sum_{l=1}^{L} \hat{Y}_{vl} = 1$, where $\mathcal{N}(v)$ is the neighbour node set of the node $v$, and $U$ is the prior distribution over all labels, $w_{u,v}$ is the edge weight between nodes $u$ and $v$, and $\mu_1$, $\mu_2$, and $\mu_3$ are hyperparameters that balance the contribution of individual terms in the objective. The terms in the objective function above encourage that: (a) the label distribution of seed nodes should be close to the ground truth, (b) the label distribution of neighbouring nodes should be similar, and, (c) if relevant, the label distribution should stay close to our prior belief. This objective function can be solved efficiently using iterative methods such as the Jacobi procedure. That is, in each step, each node aggregates the label distributions from its neighbours and adjusts its own distribution, which is then repeated until convergence. In practice, the iterative updates can be done in parallel or in a distributed fashion which then allows large graphs with a large number of nodes and labels to be trained efficiently. Bengio et al. (2006) and Ravi & Diao (2016) are good surveys on the topic for interested readers.

### 2.2 NEURAL NETWORK LEARNING

Neural networks are a class of non-linear mapping from inputs to outputs and comprised of multiple layers that can potentially learn useful representations for predicting the outputs. We will view

various models such as feedforward neural networks, recurrent neural networks and convolutional networks in the same umbrella. Given a set of $N$ training input-output pairs $\{x_n, y_n\}_{n=1}^{N}$, such neural networks are often trained by performing maximum likelihood learning, that is, tuning their parameters so that the networks' outputs are close to the ground truth under some criterion,

$$\mathcal{C}_{\text{NN}}(\theta) = \sum_n c(g_\theta(x_n), y_n), \tag{2}$$

where $g_\theta(\cdot)$ denotes the overall mapping, parameterized by $\theta$, and $c(\cdot)$ denotes a loss function such as $l$-2 for regression or cross entropy for classification. The cost function $c$ and the mapping $g$ are typically differentiable w.r.t $\theta$, which facilitates optimisation via gradient descent. Importantly, this can be scaled to a large number of training instances by employing stochastic training using minibatches of data. However, it is not clear how unlabeled data, if available, can be treated using this objective, or if extra information about the training set, such as relational structures can be used.

## 3  NEURAL GRAPH MACHINES

In this section, we devise a discriminative training objective for neural networks, that is inspired by the label propagation objective and uses both labeled and unlabeled data, and can be trained by stochastic gradient descent.

First, we take a close look at the two objective functions discussed in section 2. The label propagation objective equation 1 makes sure the predicted label distributions of neighbouring nodes to be similar, while those of labeled nodes to be close to the ground truth. For example: if a *cat* image and a *dog* image are strongly connected in a graph, and if the *cat* node is labeled as *animal*, the predicted probability of the *dog* node being *animal* is also high. In contrast, the neural network training objective equation 2 only takes into account the labeled instances, and ensure correct predictions on the training set. As a consequence, a neural network trained on the *cat* image alone will not make an accurate prediction on the *dog* image.

Such shortcoming of neural network training can be rectified by biasing the network using prior knowledge about the relationship between instances in the dataset. In particular, for the domains we are interested in, training instances (either labeled or unlabeled) that are connected in a graph, for example, *dog* and *cat* in the above example, should have similar predictions. This can be done by encouraging neighboring data points to have a similar hidden representation learnt by a neural network, resulting in a modified objective function for training neural network architectures using both labeled and unlabeled datapoints:

$$\mathcal{C}_{\text{NGM}}(\theta) = \sum_{n=1}^{V_l} c(g_\theta(x_n), y_n) + \alpha_1 \sum_{(u,v)\in\mathcal{E}_{LL}} w_{uv} d(h_\theta(x_u), h_\theta(x_v))$$
$$+ \alpha_2 \sum_{(u,v)\in\mathcal{E}_{LU}} w_{uv} d(h_\theta(x_u), h_\theta(x_v)) + \alpha_3 \sum_{(u,v)\in\mathcal{E}_{UU}} w_{uv} d(h_\theta(x_u), h_\theta(x_v)), \tag{3}$$

where $\mathcal{E}_{LL}$, $\mathcal{E}_{LU}$, and $\mathcal{E}_{UU}$ are sets of labeled-labeled, labeled-unlabeled and unlabeled-unlabeled edges correspondingly, $h(\cdot)$ represents the hidden representations of the inputs produced by the neural network, and $d(\cdot)$ is a distance metric, and $\{\alpha_1, \alpha_2, \alpha_3\}$ are hyperparameters. We call architectures to be trained using this objective *Neural Graph Machines*, and schematically illustrate the concept in figure 1. In practice, we choose an $l$-1 or $l$-2 distance metric for $d(\cdot)$, and $h(x)$ to be the last layer of the neural network. However, these choices can be changed, to a customized metric, or to using an intermediate hidden layer instead.

### 3.1  CONNECTIONS TO PREVIOUS METHODS

Note that we have separated the terms based on the edge types, as these can affect the training differently. The graph-dependent $\alpha$ hyperparameters control the balance of these terms. When $\alpha_i = 0$, the proposed objective ignores the similarity constraint and becomes a supervised-only objective as in equation 2. When $g_\theta(x) = h_\theta(x) = \hat{y}$, where $\hat{y}$ is the label distribution, the individual cost functions ($c$ and $d$) are squared $l$-2 norm, and the objective is trained using $\hat{y}$ directly instead of $\theta$, we arrive at the label propagation objective in equation 1. Therefore, the proposed objective could

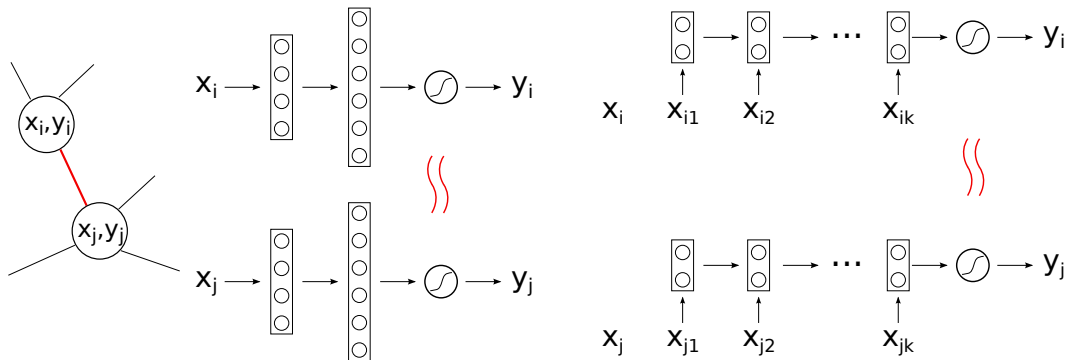

Figure 1: Illustration of Neural Graph Machine: the training objective ensures the neural net to make accurate node-level predictions and biases the hidden representations of neighbouring nodes to be similar. [Left: feedforward NNs, Right: RNNs]

be thought of as a *non-linear* version of the label propagation objective, and a *graph-regularized* version of the neural network training objective.

## 3.2 NETWORK INPUTS AND GRAPH CONSTRUCTION

Similar to graph-based label propagation, the choice of the input graphs is critical, to correctly bias the neural network's prediction. Depending on the type of the graphs and nodes on the graphs, they can be readily available to use such as social networks or protein linking networks, or they can be constructed (a) using generic graphs such as Knowledge Bases, that consists of links between vertices on the graph, (b) using embeddings learnt by an unsupervised learning technique, or, (c) using sparse feature representations for each vertex. Additionally, the proposed training objective can be easily modified for directed graphs.

We have discussed using node-level features as inputs to the neural network. In the absences of such inputs, our training scheme can still be deployed using input features derived from the graph itself. We show in figure 2 and in the experiment that the neighbourhood information such as rows in the adjacency matrix are simple to construct, yet powerful inputs to the network. These features can also be combined with existing features.

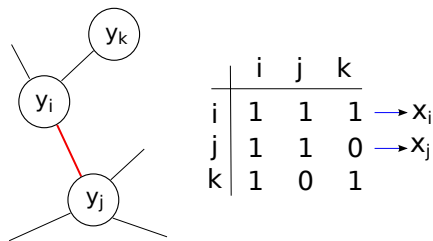

Figure 2: Illustration of how we can construct inputs to the neural network using the adjacency matrix.

## 3.3 Optimization

The proposed objective function in equation 3 has several summations over the labeled points and edges, and can be equivalently written as follows,

$$
\begin{aligned}
\mathcal{C}_{\mathrm{NGM}}(\theta) = & \sum_{(u,v)\in\mathcal{E}_{LL}} \alpha_1 w_{uv} d(h_\theta(x_u), h_\theta(x_v)) + c(g_\theta(x_u), y_u) + c(g_\theta(x_v), y_v) \\
& + \sum_{(u,v)\in\mathcal{E}_{LU}} \alpha_2 w_{uv} d(h_\theta(x_u), h_\theta(x_v)) + c(g_\theta(x_u), y_u) \\
& + \sum_{(u,v)\in\mathcal{E}_{UU}} \alpha_3 w_{uv} d(h_\theta(x_u), h_\theta(x_v)).
\end{aligned} \tag{4}
$$

The objective in its new form enables stochastic training to be deployed. In particular, in each training iteration, we use a minibatch of edges and obtain the stochastic gradients of the objective. To further reduce noise, we can select a labeled node and sample from the set of edges that are incident to that node. The number of edges per node to be sampled can be controlled.

## 3.4 Complexity

The complexity of each training epoch using equation 4 is $\mathcal{O}(M)$ where $M = |\mathcal{E}|$ is the number of edges in the graph. In practice, unlabeled-unlabeled edges do not seem to help learning and could be ignored. which further reduces the above complexity.

## 4 Experiments

In this section, we provide several experiments showing the efficacy of the proposed training objective on a wide range of tasks, datasets and network architectures. All the experiments are done using TensorFlow (Abadi et al., 2015).

## 4.1 Multi-label Classification of Nodes on Graphs

We first consider a multi-label classification on nodes of a graph. We use the *BlogCatalog* dataset (Agarwal et al., 2009), which has 10,312 nodes and 333,983 edges, and there are 39 labels. This graph represent a network of social relationships given by bloggers and the labels are the bloggers' interests. We train a feedforward neural network with one hidden layer of 50 units and train each class as a one-vs-rest binary classification task. Since there are no features for each node, we use the rows of the adjacency matrix as inputs to the network, as discussed in section 3.2. Since we use the test set to construct the graph and augment the training objective, the learning in this experiment is transductive. Since the training set is extremely unbalanced, we employ weighted sampling during training, i.e. making sure each minibatch has both positive and negative examples. In this experiment, we fix $\alpha_i$ to be equal, and experiment with $\alpha = 0$ and 0.1 (0 means no edge information during training); we use the $l$-2 metric to compute the distance between the hidden representations. We compare our method against a two-stage approach: use node2vec (Grover & Leskovec, 2016) to generate node embeddings and use a linear one-vs-rest classifier for classification. The methods are evaluated using two metrics Macro F1 and Micro F1. The results for different train/test splits and different $\alpha$ values, together with the baseline are included in table 1. The results demonstrate that 1. using the graph itself as direct inputs to the neural network and letting the network learning a non-linear mapping is more effective than the two-stage approach considered, 2. using the graph information improves the performance in the small data regime (for example: when training set is only 20% of the dataset). We observe the same improvement over Node2vec on the Micro F1 metric and $\alpha = 0.1$ is comparable to $\alpha = 0$ but performs better on the recall metric.

---

[*]These results are different compared to Grover & Leskovec (2016), since we treat the classifiers (one per label) independently. This setting is the same as for our NGM-NN classifiers.

Table 1: Results for BlogCatalog dataset averaged over 10 random splits. The higher is better.

| | Macro F1 | | |
|---|---|---|---|
| Train amount/$\alpha$ | 0 | 0.1 | Node2vec[*] |
| 0.2 | 0.180 | **0.191** | 0.168 |
| 0.5 | 0.238 | **0.242** | 0.174 |
| 0.8 | **0.263** | 0.262 | 0.177 |

## 4.2 TEXT CLASSIFICATION USING CHARACTER-LEVEL CNNS

We evaluate the proposed objective function on a multi-class text classification task using a character-level convolutional neural network (CNN). We use the AG news dataset from Zhang et al. (2015), where the task is to classify a news article into one of 4 categories. Each category has 30,000 examples for training and 1,900 examples for testing. In addition to the train and test sets, there are 111,469 examples that are treated as unlabeled examples.

We restrict the graph construction to only the train set and the unlabeled examples and keep the test set only for evaluation. We use the Google News word2vec corpus to calculate the average embedding for each news article and use the cosine similarity of document embeddings as a similarity metric. Each node is restricted to 5 neighbors.

We construct the CNN in the same way as Zhang et al. (2015), but with significantly smaller layers, as shown in table 2:

Table 2: Settings of CNNs for the text classification experiment

| Setting | Baseline "small" CNN | "Tiny" CNN |
|---|---|---|
| # of convolutional layers | 6 | 3 |
| Frame size in conv. layers | 256 | 32 |
| # of fully-connected layers | 3 | 3 |
| Hidden units in fully-connected layers | 1024 | 256 |

The network is trained with the same parameters as Zhang et al. (2015) but only for 20 epochs. We compare the final outputs using the cross entropy loss, that is $d = \text{cross\_entropy}(g(x_u), g(x_v))$. Using the proposed objective function, the NGM-CNN provides a 1.8% absolute and 2.1% relative improvement in accuracy, despite using a smaller network. We show the results in table 3.

Table 3: Results for News Categorization using CNNs

| Network | Accuracy % |
|---|---|
| Baseline: "small" CNN | 84.35 |
| Baseline: "small" CNN with thesaurus augmentation | 85.20 |
| Baseline: "tiny" CNN | 85.07 |
| "Tiny" CNN with NGM | **86.90** |

## 4.3 SEMANTIC INTENT CLASSIFICATION USING LSTM RNNS

Finally, we compare the performance of our approach for training RNN sequence models (LSTM) for a semantic intent classification task as described in the recent work on SmartReply (Kannan et al., 2016) for automatically generating short email responses. One of the underlying tasks in SmartReply is to discover and map short response messages to semantic intent clusters.[1] We choose 20 intent classes and created a dataset comprised of 5,483 samples (3,832 for training, 560 for validation and 1,091 for testing). Each sample instance corresponds to a short response message text paired with a semantic intent category that was manually verified by human annotators. For example, *"That*

---

[1]For details regarding SmartReply and how the semantic intent clusters are generated, refer Kannan et al. (2016).

*sounds awesome!"* and *"Sounds fabulous"* belong to the *sounds good* intent cluster. We construct a sparse graph in a similar manner as the news categorization task using word2vec embeddings over the message text and computing similarity to generate a response message graph with fixed node degree (k=10). We use $l$-2 for the distance metric $d(\cdot)$ and choose $\alpha$ based on the development set.

We run the experiments for a fixed number of time steps and pick the best results on the development set. A multilayer LSTM architecture (2 layers, 100 dimensions) is used for the RNN sequence model. The LSTM model and its NGM variant are also compared against other baseline systems—*Random* baseline ranks the intent categories randomly and *Frequency* baseline ranks them in order of their frequency in the training corpus. To evaluate the intent prediction quality of different approaches, for each test instance, we compute the rank of the actual intent category $rank_i$ with respect to the ranking produced by the method and use this to calculate the Mean Reciprocal Rank:

$$\text{MRR} = \frac{1}{N} \sum_{i=1}^{N} \frac{1}{\text{rank}_i}$$

We show in table 4 that LSTM RNNs with our proposed graph-augmented training objective function outperform standard baselines by offering a better MRR.

Table 4: Results for Semantic Intent Classification using LSTM RNNs

| Model | Mean Reciprocal Rank (MRR) |
|---|---|
| Random | 0.175 |
| Frequency | 0.258 |
| LSTM | 0.276 |
| NGM-LSTM | **0.284** |

## 5  CONCLUSIONS

We have proposed a training objective for neural network architectures that can leverage both labeled and unlabeled data. Inspired by the label propagation objective function, the proposed objective biases the neural networks to learn similar hidden representations for nodes connected by an edge on the graph. Importantly, this objective can be trained by stochastic gradient descent, as in supervised neural network training. We validate the efficacy of the graph-augmented objective on various state-of-the-art neural network architectures on bloggers' interest, text category and semantic intent classification problems. Additionally, the node-level input features can be combine with graph features as inputs to the neural network. We showed that a neural network that simply takes the adjacency matrix of a graph and produces node labels, can perform better than a recently proposed two-stage approach using sophisticated graph embeddings and a linear classifier.

While our objective can be applied to multiple graphs which come from different domains, we have not fully explored this aspect and leave this as future work. We expect the domain-specific networks can interact with the graphs to determine the importance of each domain/graph source in prediction. Another possible future work is to use our objective on directed graphs, that is to control the direction of influence between nodes during training.

ACKNOWLEDGMENTS

We would like to thank the Google Expander team for insightful feedbacks.

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
