# Peer review of "Neural Graph Machines: Learning Neural Networks Using Graphs"

_ICLR 2017 — rejected_

[Official Review · AnonReviewer1 · rating 3 · confidence 4 · 16 Dec 2016]

The paper proposes a model that aims at learning to label nodes of graph in a semi-supervised setting. The idea of the model is based on the use of the graph structure to regularize the representations learned at the node levels. Experimental results are provided on different tasks

The underlying idea of this paper (graph regularization) has been already explored in different papers – e.g 'Learning latent representations of nodes for classifying in heterogeneous social networks' [Jacob et al. 2014],   [Weston et al 2012] where a real graph structure is used instead of a built one. The experiments lack of strong comparisons with other graph models (e.g Iterative Classification, 'Learning from labeled and unlabeled data on a directed graph', ...). So the novelty of the paper and the experimental protocol are not strong enough to accpet the paper.

Pros:
* Learning over graph is an important topic

Cons:
* Many existing approaches have already exploited the same types of ideas, resulting in very close models
* Lack of comparison w.r.t existing models

[Official Review · AnonReviewer3 · rating 4 · confidence 4 · 17 Dec 2016]
**Very similar to previous work.**

This paper proposes the Neural Graph Machine that adds in graph regularization on neural network hidden representations to improve network learning and take the graph structure into account.  The proposed model, however, is almost identical to that of Weston et al. 2012.

As the authors have clarified in the answers to the questions, there are a few new things that previous work did not do:

1. they showed that graph augmented training for a range of different types of networks, including FF, CNN, RNNs etc. and works on a range of problems.
2. graphs help to train better networks, e.g. 3 layer CNN with graphs does as well as than 9 layer CNNs
3. graph augmented training works on a variety of different kinds of graphs.

However, all these points mentioned above seems to simply be different applications of the graph augmented training idea, and observations made during the applications.  I think it is therefore not proper to call the proposed model a novel model with a new name Neural Graph Machine, but rather making it clear in the paper that this is an empirical study of the model proposed by Weston et al. 2012 to different problems would be more acceptable.

[Official Review · AnonReviewer2 · rating 3 · confidence 4 · 18 Dec 2016]
**Very similar to previous work, rebranded.**

The authors introduce a semi-supervised method for neural networks, inspired from label propagation.

The method appears to be exactly the same than the one proposed in (Weston et al, 2008) (the authors cite the 2012 paper). The optimized objective function in eq (4) is exactly the same than eq (9) in (Weston et al, 2008).

As possible novelty, the authors propose to use the adjacency matrix as input to the neural network, when there are no other features, and show success on the BlogCatalog dataset.

Experiments on text classification use neighbors according to word2vec average embedding to build the adjacency matrix. Top reported accuracies are not convincing compared to (Zhang et al, 2015) reported performance. Last experiment is on semantic intent classification, which a custom dataset; neighbors are also found according to a word2vec metric.

In summary, the paper propose few applications to the original (Weston et al, 2008) paper. It rebrands the algorithm under a new name, and does not bring any scientific novelty, and the experimental section lacks existing baselines to be convincing.

[Author Response · Sujith Ravi · 21 Jan 2017]
**AC/Reviewer response**

We thank the reviewers for all their comments. However we would like to respond, make further clarifications and show new results (see comment #3 below) that should address all reviewer concerns.

1. Response to all reviewers

This work generalizes the Weston et al.’s work on semi-supervised embedding and extends it to new settings (for example, when only graph inputs & no features are available). Unlike the previous works, we show that the graph augmented training method can work with multiple neural network architectures (Feed Forward NNs, CNNs, RNNs) and on multiple prediction tasks & datasets using "natural" as well as "constructed" graphs. The experiment results clearly show the effectiveness of this method in all these different settings. Besides the methodology, our study also presents an important contribution towards assessing the effectiveness of graph+neural networks as a generic training mechanism for different architectures & problems, which was not well studied in previous papers. We can add more clarifications in our paper to emphasize this point more clearly. 

Furthermore, the reviewers’ concerns should also be addressed with details in comment #3 (please see below), where we show new experiments and direct comparison results against Weston et al., 2012; Yang et al., 2016 [2] and other graph methods. 

2. Regarding reviewer #2’s comment about accuracy comparison against Zhang et al., 2015.

We would like to make it clear to the reviewers that the numbers reported on this task (Table 3 in our paper) for the “small CNN” baseline were obtained from their paper (Zhang et al., 2015 refers to this as “Small ConvNet” which uses 6 layers). They use far deeper and more complex networks (wrt frame size, hidden units) to achieve the reported accuracies on this task.
In comparison, NGM trained with a simple 3-layer CNN achieves much better results (86.90 vs 84.35/85.20) and also produces comparable results to their best 9-layer Large ConvNet. This shows that our graph augmented training method not only achieves better results but also helps train more efficient, compressible networks.

3. Response to all reviewers — regarding comparison against Weston et al., 2012; Yang et al., 2016 [2] and other graph methods. This, along with experiments already detailed in the paper, should address all the concerns raised about how the new method performs in comparison to previous works.

We performed new experiments and compare our method on a new task with very limited supervision — the PubMed document classification task [1]. The task is to classify each document into one of 3 classes, with each document being described by a TF-IDF weighted word vector. The graph is available as a citation network: two documents are connected to each other if one cites the other. The graph has 19,717 nodes and 44,338 edges, with each class having 20 seed nodes and 1000 test nodes. In our experiments we exclude the test nodes from the graph entirely, training only on the labeled and unlabeled nodes.

We train a feed-forward network with two hidden layers with 250 and 100 neurons, using the l2 distance metric on the last hidden layer. The NGM model is trained with alpha_i = 0.2, while the baseline is trained with alpha_i = 0 (i.e., a supervised-only model). We use self-training to train the model, starting with just the 60 seed nodes as training data. The amount of  training data is iteratively increased by assigning labels to the immediate neighbors of the labeled nodes, and retraining the model.

We compare the final NGM model against the baseline, the Planetoid models (Yang et. al, 2016 [2]), semi-supervised embedding (SemiEmb) (Weston et. al, 2012), manifold regression (ManiReg) (Belkin et. al, 2006), transductive SVM (TSVM), label propagation (LP) and other methods. The results show that the NGM model (without any “tuning”) outperforms the baseline, semi-supervised embedding, manifold regularization and Planetoid-G/Planetoid-T, and compares favorably with the other Planetoid system. We believe that with tuning, NGM accuracy can be improved even further.


Table: Results for Pubmed document classification

Feat*              0.698
SemiEmb*      0.711 
ManiReg*       0.707
 Planetoid-I*    0.772
TSVM*            0.622 
LP*                 0.630
 GraphEmb*    0.653 
Planetoid-G*  0.664 
Planetoid-T*   0.757

 ------------------------------------ 
Feed forward NN (similar to Feat)   0.709 
NGM           0.759

*numbers reported in Table 3 from Yang et al., 2016 


[1]: Sen, Prithviraj, Namata, Galileo, Bilgic, Mustafa, Getoor, Lise, Galligher, Brian, and Eliassi-Rad, Tina. Collective classification in network data. AI magazine, 29(3):93, 2008.
[2]: Yang, Zhilin, Cohen, William, and Salakhutdinov, Ruslan. Revisiting Semi-Supervised Learning with Graph Embeddings., 2016.

[Final Decision · Program Chairs · 06 Feb 2017]
**ICLR committee final decision**

The paper is an interesting contribution, primarily in its generalization of Weston's et al's work on semi-supervised embedding method. You have shown convincingly that it can work with multiple architectures, and with various forms of graph. And the PubMed results are good. To improve the paper in the future, I'd recommend 1) relating better to prior work, and 2) extending your exploration of its application to graphs without features.